# Fragile X-Associated Neuropsychiatric Disorders (FXAND) in Young Fragile X Premutation Carriers

**DOI:** 10.3390/genes13122399

**Published:** 2022-12-17

**Authors:** Ramkumar Aishworiya, Dragana Protic, Si Jie Tang, Andrea Schneider, Flora Tassone, Randi Hagerman

**Affiliations:** 1Medical Investigation of Neurodevelopmental Disorders (MIND) Institute, University of California Davis, 2825 50th Street, Sacramento, CA 95817, USA; 2Khoo Teck Puat-National University Children’s Medical Institute, National University Health System, 5 Lower Kent Ridge Road, Singapore 119074, Singapore; 3Department of Pediatrics, Yong Loo Lin School of Medicine, National University of Singapore, 10 Medical Drive, Singapore 117597, Singapore; 4Department of Pharmacology, Clinical Pharmacology and Toxicology, Faculty of Medicine, University of Belgrade, 11000 Belgrade, Serbia; 5Department of Pediatrics, School of Medicine, University of California Davis, 4610 X St, Sacramento, CA 95817, USA; 6Department of Biochemistry and Molecular Medicine, School of Medicine, University of California Davis, 4610 X St, Sacramento, CA 95817, USA

**Keywords:** Fragile X premutation, adolescent, anxiety, ASD, FSIQ, *FMR1*

## Abstract

**Background:** The fragile X premutation carrier state (PM) (55–200 CGG repeats in the fragile X messenger ribonucleoprotein 1, *FMR1* gene) is associated with several conditions, including fragile X-associated primary ovarian insufficiency (FXPOI) and fragile X-associated tremor ataxia (FXTAS), with current literature largely primarily investigating older PM individuals. The aim of this study was to identify the prevalence of fragile X-associated neurodevelopmental disorders (FXAND) in a sample of young PM individuals. **Methods:** This was a retrospective study conducted through a medical record review of PM individuals who were seen either for clinical concerns (probands, 45.9%) or identified through the cascade testing (non-probands, 54.1%) of an affected sibling with fragile X syndrome. Information on the presence of autism spectrum disorder, attention deficit hyperactivity disorder, anxiety, depression, long-term psychiatric medication intake, and cognitive function, based on standardized assessments, was obtained. Molecular data, including CGG repeat number and *FMR1* mRNA levels, were also available for a subset of participants. Analysis included descriptive statistics and a test of comparison to describe the clinical profile of PM individuals pertinent to FXAND. **Results:** Participants included 61 individuals (52 males and 9 females) aged 7.8 to 20.0 years (mean 12.6 ± 3.4) with a mean full-scale IQ of 90.9 ± 22.7. The majority (N = 52; 85.2%) had at least one mental health disorder, with anxiety being the most common (82.0% of subjects), followed by ADHD (66.5%), and ASD (32.8%). Twenty-seven (87.1%) of non-probands also had at least one mental health condition, with probands having lower cognitive and adaptive skills than non-probands. ASD was present in 20 participants (17/52 males and 3/9 females; 15 probands) with significantly lower FSIQ in those with ASD (mean 73.5 vs. 98.0, *p* < 0.001). Participants with ASD had a higher number of long-term medications compared to those without (2.32 vs. 1.3, *p* = 0.002). **Conclusions:** Our findings indicate a high rate of FXAND diagnoses within a cohort of young PM individuals, including those identified via cascade testing, although this was not a population sample. An awareness of the entity of FXAND and the early recognition of the symptoms of associated conditions may facilitate timely and appropriate care for PM individuals.

## 1. Background

While the clinical entity of fragile X syndrome (FXS) has been well defined and characterized over the past four decades, the associated phenotypes of the fragile X premutation (PM) is relatively newly recognized, since the early 2000s. FXS is caused by a full expansion of the CGG repeats (≥200 repeats) in the 5′ region of the fragile X messenger ribonucleoprotein 1 (*FMR1)* gene, resulting in the loss of function of the gene and the consequent deficiency or absence of the *FMR1* protein (FMRP) [1]. FXS is the most common monogenetic cause of intellectual disability (ID) and autism spectrum disorder (ASD), with an estimated prevalence of 1 in 5000 males and 1 in 8000 females, and hallmark clinical features of ID, facial dysmorphism, language delays, and aggressive behavior [2,3,4]. In turn, the PM state refers to individuals who have between 55 and 200 CGG repeats, and whereby, the progeny of females can have a full expansion of the CGG repeats to result in FXS [5]. The PM state is more common than the full mutation, occurring in 1 in 200 women and in 1 in 400 men [2]. Several conditions have been associated with the PM state, including fragile X-associated primary ovarian insufficiency (FXPOI) [6,7] and fragile X-associated tremor ataxia (FXTAS) [8,9,10,11]. The pathophysiology of these conditions is attributed with the RNA toxicity caused by elevated levels of *FMR1* mRNA in PM individuals [12].

Recently, a newly recognized implication of the PM state was labeled in 2018 as fragile X-associated neurodevelopmental disorders (FXAND) [13]. The most common conditions under this umbrella include anxiety and depression, but several other neurodevelopmental disorders, including phobias and obsessive-compulsive disorders, have also been reported as being part of this entity, with a higher incidence occurring in PM individuals compared to controls without the premutation [14,15,16]. However, these initial studies have primarily focused on adult PM individuals to profile their FXAND conditions in adulthood. The literature is currently sparse on the clinical profile of young PM individuals, and the presence and onset (if applicable) of FXAND in the early years of life. Of note, there is also a growing body of literature highlighting the increased occurrence of ASD and attention-deficit hyperactivity disorder (ADHD) in PM individuals, which would fall under FXAND [17,18,19]. Previous research studies have made the distinction between PM individuals referred to for clinical concerns of FXAND disorders, and those with no apparent clinical concerns but that have been identified through the cascade testing of a sibling with FXS or the premutation. The rates of FXAND disorders are typically higher in those with clinical concerns compared to those included through cascade testing. Among those identified through cascade testing, the rates tend to be either similar to or slightly increased to the population prevalence rates of these conditions [18,20]. Given that these are primarily childhood-based neuropsychiatric disorders that can respond to treatment, it would be worthwhile to examine this further in order to study FXAND and its associations in greater detail. 

Hence in this study, our primary goal is to utilize a cohort of young PM individuals to assess the presence, relationship, and implications of various FXAND-related conditions (including ASD, ADHD, anxiety, and depression). Our secondary aim is to identify the clinical correlates of the presence of these disorders. 

## 2. Methods

This was a retrospective cross-sectional study conducted through a medical record review of young PM individuals. Institutional review board (IRB) approval was obtained for all study-related activities from the UC Davis IRB (study number 254134). 

### 2.1. Subjects

PM individuals, between 8 and 21 years of age inclusive, who were either non-probands identified through cascade testing (the routine genetic testing of family members after the identification of a proband) of an affected sibling with FXS (N = 33, 54.1%) or probands who were seen for clinical concerns at the Fragile X Treatment and Research Center at the University of California Davis MIND Institute, a tertiary healthcare center, between 2004 and 2022 (N = 28, 45.9%) were included. All study participants had the documentation of their PM state confirmed via the detection of a premutation allele through DNA diagnostic testing, which included both polymerase chain reaction and southern blot analysis. 

### 2.2. Study Measures

#### 2.2.1. Clinical Phenotype-Related Measures

Clinical information about the PM participants was obtained using a structured data abstraction form, and included demographic data and details on the presence and severity of conditions of interest, including ASD, ADHD, anxiety, depression, obsessive compulsive disorder, specific phobias, and panic disorders (referred to as FXAND-related conditions). The presence of these conditions were determined by licensed and research-reliable healthcare providers in accordance with the Diagnostic and Statistical Manual—5th Edition (DSM-5) criteria. In addition, the diagnosis of ASD was made using the Autism Diagnostic Observation Schedule (ADOS) [21]. Data related to ASD diagnoses were available for 50 participants. Information on cognitive assessment based on standardized testing was also obtained. Given the varying age range of the participants, a variety of standardized cognitive assessments were used across the study participants. This included the Wechsler Intelligence Scale for Children, Third or Fourth Edition (WISC-III or WISC-IV) [22], the Stanford Binet Intelligence Scales, Fifth Edition (SB-5) [23], and the Wechsler Adult Intelligence Scale, Third or Fourth Edition (WAIS-III or WAIS-IV) [24], as per participant age and clinical profile. Adaptive functioning was assessed using the Vineland Adaptive Behavior Scales—Second Edition (VABS–II) [25], which provides an adaptive behavior composite score (ABC) and sub-domain scores in Communication, Socialization and Daily Living Skills. The presence of mental health disorders was made using the Structured Clinical Interview for DSM-IV Axis I Personality Disorders (SCID) [26]. Information on long-term psychiatric medication intake was also obtained; this included but was not limited to medications such as antidepressants, antipsychotics, and stimulant groups of medications. Medications in other related classes/fields of medicine such as anti-epileptics were also documented.

#### 2.2.2. Molecular Data

Molecular data (CGG repeat number, and *FMR1* mRNA levels) were also available for a subset of the study sample, and were recorded for analysis. Their PM state was previously confirmed and determined using Southern Blot and PCR-based CGG repeat genotyping on genomic DNA isolated from whole blood. The details are as described in Tassone et al., 2008, and Filipovick et al., 2010 [27,28]. The expression levels of *FMR1* mRNA and *FMR1* protein (FMRP) were measured as previously described (Tassone et al., 2000) [12]. Data related to the *FMR1* mRNA and FMRP levels were available for 47 and 15 participants, respectively.

#### 2.2.3. Statistical Analysis

Statistical analysis was performed using the Statistical Package for Social Sciences (SPSS) for Windows [29]. Descriptive statistics were used to examine the demographic and clinical variables of interest, including various FXAND conditions. The results were expressed as mean ± standard deviation of mean. Tests of comparison (two sample *t*-tests, Fisher’s exact, and Chi-squared tests) were used to examine for differences in clinical phenotype based on the ages of the subjects, number of CGG repeats, and the presence of the main FXAND conditions of anxiety, ASD, and depression. All significance values were set to *p* < 0.05. Figures were created on GraphPad Prism Version 9.

## 3. Results

A total of 61 participants (52 males and 9 females) were included in the final analysis. The mean age of the whole sample was 12.6 years (SD 3.4; median 12.0, age range: 7.8–20.0 years). Table 1 shows the descriptive demographic, clinical, and molecular data of all study participants. Females were older and had a lower CGG repeat allele size, as compared to males. There were no other significant differences in terms of gender, except for FMRP levels, which were higher in females as compared to males (however, these data were available only for 15 participants, 7 males, and 8 females). The mean full-scale IQ (FSIQ) of the cohort was 90.9 (SD 22.7), which is within the average range, as expected; there were no differences in IQ profile between males and females. The mean number of CGG repeats was 96.7 (SD 41.2); 21 subjects had >100 CGG repeats. Comparing probands and non-probands, there were no statistically significant differences in terms of gender, age, and molecular data (CGG repeats, FMRP levels, and mRNA levels). However, non-probands had significantly higher FSIQ (99.3 ± 16.6) scores as compared to probands (79.0 ± 25.0; *p* = 0.001), and higher adaptive behavior scores as well (VABS ABC 89.8 ± 9.1 vs. 71.9 ± 14.7, *p* = 0.004). Scores on the performance IQ (PIQ), verbal IQ (VIQ), VABS Socialization and Daily Living domains were also higher in non-probands as compared to probands (Table 1).

## 4. FXAND-Related Conditions

Figure 1A and Table 2 present data regarding FXAND-related conditions in the sample of young PM carriers. The majority (57/61, 93.4%) of participants had at least one FXAND-related condition (Figure 1A), with anxiety being the most common (total: 50/61, and 43/52 males and 7/9 females) and ASD being present in 20 participants (17/52 males and 3/9 females). ADHD occurred more frequently in males as compared to females (39/52 vs. 1/9, *p* < 0.001; Table 2); there were no other differences based on gender. There was no difference between the average number of FXAND-related conditions between probands and non-probands (2.9 ± 1.3 vs. 2.4 ± 1.3, *p* = 0.20, Figure 1B). Even among non-probands, 30/33 participants had at least one FXAND-related condition. Non-probands were less likely to have ASD as compared to those presenting clinically (5/25 vs. 15/25, χ^2^ = 8.3, *p* = 0.04; Table 2). However, the other FXAND conditions were equally likely to occur in both participants identified via cascade testing and via clinical concerns. The number of CGG repeats or *FMR1* mRNA levels were not significantly associated with the presence of any FXAND conditions (*p* > 0.05)). 

### 4.1. Factors Associated with Clinical Phenotype Features

The presence of ASD was significantly correlated with IQ, with those having ASD having lower FSIQ (73.5 ± 21.7 vs. 98.0 ± 20.0, *p* < 0.001). This applied to verbal and non-verbal IQ as well, as shown in Table 3. Those with a diagnosis of ASD also had a higher number of long-term medications being prescribed, compared to those without (2.32 ± 1.0 vs. 1.3 ± 1.11, *p* = 0.002). The presence of any other FXAND condition was not significantly associated with cognition or medication use. Probands with ASD had significantly lower FSIQ scores than non-probands with ASD (62.1 ± 14.6 vs. 94.0 ± 16.8, *p* = 0.003). The VIQ, PIQ, and VABS daily living scores were also lower for probands with ASD, compared to non-probands with ASD (Table 3).

### 4.2. Psychotropic Medication Intake 

Table 4 details data pertinent to long-term psychotropic medication use by the study sample. The majority (57/61, 93.4%) were on some form of long-term medications, with common categories of medication being stimulants and selective serotonin reuptake inhibitors (SSRIs). This was not surprising, given that anxiety and ADHD were frequent within the sample. Nearly half of the sample (28/61, 45.9%) were on two or more medications, with the range varying from 0 to 5 medications. There was no difference in the types of medications being used by males and females (Table 4). Probands more frequently used antipsychotics than non-probands (35.7% vs. 6.1%, *p* = 0.005, Table 4). There were no differences in the usage of other psychiatric drugs between probands and non-probands.

## 5. Discussion

In this study, we demonstrate that FXAND-related conditions among young individuals (probands and non-probands) with the PM state occurred frequently, with anxiety and ADHD being the most common. Further, the presence of ASD in the study’s cohort was associated with lower cognitive skills, and similarly, probands with ASD had significantly lower IQ than non-probands with ASD. Finally, the presence of ASD was associated with a greater number of long-term medication intake, with SSRIs being the most frequently used drugs in this cohort. Interestingly, antipsychotic medications were more frequently used in probands than in non-probands. 

Our results with respect to FXAND conditions in young individuals with PM add to the existing literature by showing the occurrence of these conditions in as young as early childhood. Consistent with previous similar studies, more than four-fifths of our included participants had anxiety. The incidence of anxiety in this cohort was similar to the study by Cordeiro et al., which also looked at young premutation carriers and described the incidence of anxiety in more than two-thirds of that study’s participants [30]. These rates are significantly higher than the general population prevalence rates (2.4–10.7%) in children and young adults [31,32,33]. Similarly rates of ADHD and depression in this cohort are much higher than that in the general population, which is typically at around 9.8% and 4.4%, respectively [32,33].

In particular, the common neurodevelopmental disorders of ASD and ADHD occurred frequently in this current study’s cohort. Indeed, the incidence of ASD was much higher in this cohort (32.8%) as compared to the 1.0–2.3% prevalence rates in the general population [34,35]. Previous studies have also highlighted higher rates of ASD and ASD traits in PM carriers as compared to the general population [17,18,36]. For example, Farzin and colleagues (2006) suggested that PM carriers, even those who do not present clinically, may be at increased risk for ASD and/or ADHD, based on a small sample size (14 male probands who presented to clinic, 13 male non-probands, and control group of 16 male siblings) [18]. Our findings of ASD being more common in probands than in non-probands is also in agreement with this study and also with the one by Chonchaiva et al. (2011) [18,20]. In addition, polypharmacy is known to be higher in individuals with ASD as compared to the general population, and this is likely due to multiple factors, including a higher incidence of comorbidities and challenging behaviors, necessitating multiple medications [37,38]. PM carriers may also be at a higher risk of having additional genetic hits that may contribute to the presence of ASD, exacerbating cognitive deficits [39,40]. The intrinsic vulnerability of the PM state to environmental toxins such as alcohol and pesticides may potentially be a part of this relationship with the presence of FXAND disorders, including ASD [40].

Why some young individuals with the PM have ASD is thought to be associated with several molecular factors. It could be related to the elevated levels of *FMR1* mRNA in carriers, which leads to toxicity via calcium dysregulation, mitochondrial dysfunction, and DNA damage repair functions or other mechanisms [12,41,42,43]. This disruption of normal molecular function is believed to occur in neurons and astrocytes throughout the brain, and has been documented in individuals with FXTAS, which is a later stage disease in older PM individuals [42,44,45]. Other possible underlying mechanisms of the presence of ASD in these individuals could include mitochondrial dysfunction, and reduced levels of FMRP, which can dysregulate the translation of other genes associated with ASD [40,46,47]. Our findings also lend weight to the increased comorbidity associated with ASD in terms of reduced cognitive scores, given the known occurrence of intellectual disability in 30–50% of individuals with ASD [48,49].

Our findings also show that PM carriers as a whole have normal cognition and adaptive behavior scores. This was the case for non-probands who did not have any clinical concerns and who were identified through cascade testing. Given that probands were PM carriers who had clinical concerns and hence sought treatment at the center, it is likely that they had some form of impairment in their functioning; this could be related to the presence of ASD, which occurred more commonly in probands as well. This would explain the lower IQ and adaptive behavior scores in probands compared to non-probands. However, even among non-probands with normal cognition, we saw a substantial proportion with FXAND conditions. The clinical implications of our results include the need for clinical providers and personnel working with PM carriers to be aware of the existence and implications of FXAND-related conditions. The monitoring of PM siblings of children with FXS may also be useful. This will facilitate appropriate recognition, diagnosis, and intervention for these conditions, if and when they arise in early childhood. Specifically, given the high rates of anxiety within our cohort, a low threshold for further evaluations of PM individuals experiencing symptoms of anxiety would be recommended to allow for the identification and treatment of anxiety, as applicable.

This study has several limitations. First, we recognize the potential for sample bias, as this was not a community/population-based sample. Hence, the results should be interpreted with caution and will not be equivalent to population-level prevalence values. Secondly, the relatively small sample size led to reduced statistical power for comparative analysis between groups. Future studies will incorporate a larger cohort of PM carriers to examine their clinical phenotype further.

## 6. Conclusions

The Fragile X premutation state is associated with several neurodevelopmental disorders under the term FXAND, and these are present, even in childhood and young adulthood. Clinical providers need to be aware of the entity of FXAND when counselling and evaluating young PM individuals. The early identification and appropriate treatment of these conditions will be crucial for long-term mental health and overall outcomes.

## Figures and Tables

**Figure 1 genes-13-02399-f001:**
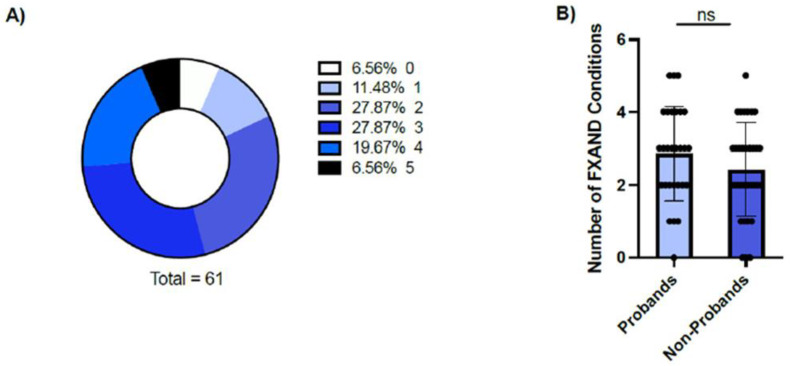
(**A**) Percentage of patients with 0, 1, 2, 3, 4 and 5 FXAND conditions. (**B**) Mean number of FXAND conditions in Probands and Non-Probands. Data mean ± SD. *p* = 0.1950.

**Table 1 genes-13-02399-t001:** Descriptive data of study subjects.

Variable	All	Males	Females	*p*-Value	Probands	Non-Probands	*p*-Value
N (%)	61	52 (85.2)	9 (14.5)	/	28	33	/
Males (%)	/	/	/	/	24 (86)	28 (85)	1.0
Age range (years)	7.8–20.0	8–20	9–19	/	8–19	8–20	/
Mean age (years)	12.6 ± 3.4	12.2 ± 3.3	14.7 ± 3.5	**0.04**	12.8 ± 3.6	12.2 ± 3.2	0.47
	Mean (SD)						
CGG repeat size	96.7 ± 41.2	101.0 ± 42.4	73.0 ± 24.0	**0.01**	98.5 ± 46.3	95.0 ± 36.9	0.75
Mean mRNA level (AU)	2.8 ± 1.1 ^†^	2.9 ± 1.1 ^†^	1.6 ± 0	NA	2.9 ± 1.2	2.8 ± 1.1	0.76
Mean FMRP level (AU)	1.1 ± 0.4	0.9 ± 0.4 ^‡^	1.3 ± 0.3 ^‡^	**0.04**	1.1 ± 0.3	1.1 ± 0.4	1.00
FSIQ	90.9 ± 22.7	91.2 ± 21.8	89.3 ± 28.1	0.83	79.0 ± 25.0	99.3 ± 16.6	**0.001**
PIQ	93.0 ± 20.9	91.1 ± 20.9	98.6 ± 21.5	0.42	78.6 ± 21.1	101.5 ± 16.0	**0.004**
VIQ	96.5 ± 25.2	95.7 ± 27.1	98.7 ± 20.4	0.79	83.3 ± 32.5	103.8 ± 17.1	**0.04**
VABS ABC	81.8 ± 14.7	81.1 ± 15.4	88.0 ± 3.6	0.54	71.9 ± 14.7	89.8 ± 9.1	**0.004**
VABS Communication	79.5.5 ± 15.7	79.4 ± 16.1	81.0 ± 17.0	0.89	72.8 ± 11.8	85.6 ± 16.9	0.0750
VABS Socialization	86.1 ± 16.3	84.1 ± 16.1	102.0 ± 2.8	0.15	75.0 ± 17.6	94.9 ± 7.9	**0.005**
VABS Daily living	85.8 ± 17.8	85.1 ± 18.7	91.5 ± 6.4	0.64	74.8 ± 18.1	95.7 ± 10.6	**0.006**

Abbreviations: FMRP = *FMR1* protein, FSIQ = Full Scale IQ, PIQ = Performance IQ, VIQ = Verbal IQ, VABS = Vineland Adaptive Behavior Scales, ABC = Adaptive Behavior Composite; ^†^ The mRNA level was obtained for 46 males and 1 female; ^‡^ The FMRP level was obtained for 7 males and 8 females. Statistically significant *p* values are bolded.

**Table 2 genes-13-02399-t002:** FXAND-related conditions in the samples of young PM carriers.

Condition	All (N = 61)	Male (N = 52)	Female (N = 9)	*p*-Value	Probands (N = 28)	Non-Probands (N = 33)	*p*-Value
ASD (N, %)	20 (32.8)	17 (32.7)	3 (33.3)	1	15 (53.6)	5 (15.2)	**0.04**
ADHD (N, %)	40 (65.6)	39 (97.5)	1 (11.1)	**0.0004**	18 (64.3)	22 (66.7)	0.37
OCD (N, %)	20 (32.8)	19 (36.5)	1 (11.1)	0.24	11 (39.3)	9 (27.3)	0.27
Anxiety (N, %)	50 (82.0)	43 (82.7)	7 (77.8)	1	23 (82.1)	27 (81.2)	0.38
Social Phobia (N, %)	14 (23.0)	13 (25.0)	1 (11.1)	0.44	6 (21.4)	8 (24.2)	0.86
Depression (N, %)	12 (19.7)	10 (19.2)	2 (22.2)	1	5 (17.9)	7 (21.2)	0.85

Abbreviations: ASD = autism spectrum disorder, ADHD = attention deficit hyperactivity disorder, OCD = obsessive compulsive disorder; Fisher Exact Probability Test was used due to numbers in some cells being less than 5. Statistically significant *p* values are bolded.

**Table 3 genes-13-02399-t003:** Presence of ASD in young PM carriers and its associated variables.

Variables	ASD Present (N = 20)	ASD Absent (N = 30)	*p*-Value	Probands with ASD (N = 15)	Non-Probands with ASD (N = 5)	*p*-Value
	Mean (SD)					
Age in years	12.7 (+3.5)	12.3 (3.2)	0.60	14.0 (3.2)	9.2 (1.34)	**0.005**
CGG repeat size	95.9 (46.4)	96.1 (39.3)	0.99	100 (47.8)	81.5 (44.0)	0.50
mRNA level (AU)	2.8 (1.3)	2.8 (1.1)	0.87	3.1 (1.4)	2.1 (0.4)	0.19
FSIQ	73.5 (21.7)	98.0 (20.0)	**0.001**	62.1 (14.6)	94.0 (16.8)	**0.003**
VIQ	75.1 (20.4)	102.9 (23.2)	**0.009**	64.2 (11.3)	93.3 (20.1)	**0.04**
PIQ	77.6 (16.3)	101.0 (20.2)	**0.01**	70.2 (11.3)	96.0 (11.3)	**0.04**
VABS ABC	76.8 (15.0)	86.7 (13.4)	0.14	69.7 (14.3)	87.5 (9.0)	0.06
VABS Communication	76.6 (15.1)	82.8 (16.7)	0.41	71.2 (12.3)	84.8 (16.8)	0.17
VABS Socialization	81.0 (17.2)	91.1 (14.4)	0.12	73.0 (19.1)	91.0 (7.8)	0.12
VABS Daily living	82.0 (19.9)	90.0 (15.0)	0.34	72.3 (17.0)	96.5 (15.7)	**0.05**
Number of prescribed medications	2.32 (1.0)	1.3 (1.11)	**0.002**	2.3 (1.1)	2.4 (0.9)	0.83
Number of Males (%)	17 (85.0)	27 (90.0)	0.59	12 (80)	5 (100)	0.54
Number with ADHD (%)	14 (70.0)	19 (63.3)	0.15	9 (60)	5 (100)	0.26
Number with Depression (%)	1 (5.0)	9 (30.0)	0.17	1 (7)	0 (0)	>0.99
Number with Anxiety (%)	16 (80.0)	24 (80.0)	0.89	12 (80)	4 (80)	>0.99

Abbreviations: FSIQ = Full-scale IQ, VIQ = Verbal IQ, PIQ = Performance IQ, VABS = Vineland Adaptive Behavior Scales, ABC = Adaptive Behavior Composite. Statistically significant *p* values are bolded.

**Table 4 genes-13-02399-t004:** Psychotropic medication intake in the samples of young PM carriers.

Intake of Medications (Yes/No)	All (N = 61)	Males (N = 52)	Female (N = 9)	*p*-Value	Probands (N = 28)	Non-Probands (N = 33)	*p*-Value
	N (%)						
Antipsychotics	12 (19.7)	12 (23.1)	0 (0)	NA	10 (35.7)	2 (6.1)	**0.005**
Antidepressants (All)	45 (75.0)	40 (76.9)	5 (55.5)	0.57	20 (71.4)	25 (75.7)	0.69
SSRIs	42 (68.9)	36 (69.2)	6 (66.7)	0.55	18 (64.28)	24 (72.7)	0.48
Stimulants	20 (32.8)	19 (36.5)	1 (11.1)	NA	8 (28.57)	12 (36.4)	0.52
α agonists	5 (8.2)	4 (7.7)	1 (11.1)	NA	2 (7.14)	3 (9.1)	1 ^†^
Melatonin	2 (3.3)	2 (3.8)	0 (0)	NA	0 (0)	2 (6.1)	0.49 ^†^
Other Medications	10 (16.4)	6 (11.5)	4 (44.4)	0.60	8 (28.57)	2 (6.1)	**0.03**

Abbreviations: SSRI: Selective serotonin reuptake inhibitors; Other Medications: anti-epileptics and minocycline. ^†^ —Fisher’s Exact Probability Test. Statistically significant *p* values are bolded.

## Data Availability

The datasets used and/or analyzed during the current study are available from the corresponding author on reasonable request.

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
