# Peer review of "Fragile X-Associated Neuropsychiatric Disorders (FXAND) in Young Fragile X Premutation Carriers"

_genes, 2022, doi:10.3390/genes13122399_

Round 1
Reviewer 1 Report
The authors present a well-conducted study on FXAND in young PM carriers. The manuscript is well structured and easy to read with a cleary stated aim and a relevant discussion of the findings and the study limitations.
The subject is relevant and could have implications for the clinical management of for instance siblings to children diagnossed with FXS.
As such, I have no major concerns regarding this manuscript.
If possible, however, I would like the authors to elaborate a little more on a few issues:
- The authors have seperated the PM carriers in probands and non-probands. Is such an distinction relevant given the relatively equal incidence of FXAND, also when looking to expand the dataset in the future?
- If accepting PM in 1:300 and anxiety rates at around 5 % in the general young population, the PM carriers account for as much of 1 in 20 of youngsters with anxiety (if my math is sound - apologies if not). In the eyes of the authors, should anxiety in a young person indicate testing for PM?
- The authors state that FMRP levels in the participants were higher in males. How does this correspond with normal physiology?
- The included participants are mostly male. Were females evenly distributed among probands and non-probands?
- The statistical comparison of males and females are somewhat difficult to interpret due to the low number of female participants. Do the authors feel that maybe FXAND should be more pronounced in boys compared with girls or is there truely no effect of sex?
- Is there any association to number of CGG repeats/FMRP levels - for instance regarding medications or number of FXAND diagnoses or severity of individual diagnoses (proband versus non-proband)?
- Do the authors have any data on any effects of mosaicism?
Author Response
Point 1: The authors have separated the PM carriers in probands and non-probands. Is such an distinction relevant given the relatively equal incidence of FXAND, also when looking to expand the dataset in the future?
Response 1: This was done to address the possible bias in analyzing the results as a whole as a higher incidence of FXAND conditions could have been driven by probands who presented with clinical concerns to the medical institution. Hence, we intentionally separated the sample into probands and non-probands, to allow for analysis of non-probands (who were well, with no clinical concerns and identified by cascade testing) separately. Interestingly, even in non-probands there was a significant incidence of FXAND conditions. This can then justify future datasets to be analyzed as a whole without the separation.
Point 2: If accepting PM in 1:300 and anxiety rates at around 5 % in the general young population, the PM carriers account for as much of 1 in 20 of youngsters with anxiety (if my math is sound - apologies if not). In the eyes of the authors, should anxiety in a young person indicate testing for PM?
Response 2: While the incidence of anxiety in this study sample of PM carriers was high, this is a single study with a relatively small sample. Hence this cannot justify the testing for the PM state in all young person presenting with anxiety. It would still be important to clincially evaluate the young person for the presence of other features of the PM state such as family history of primary ovarian insufficiency, family history of intellectual disability and fragile X syndrome. In the presence of such features and anxiety, testing for PM state can be considered.
Point 3: The authors state that FMRP levels in the participants were higher in males. How does this correspond with normal physiology?
Response 3: Within the cohort, only a small number (15) individuals had FMRP levels available. Here, the FMRP levels were higher in females as compared to males as shown in Table 1. We apologize for this error and have rectified this in the revised manuscript. While it is known that many premutation carriers have higher levels of FMRP than non-premutation carriers, gender differences in FMRP levels within premutation carriers have not been established thus far. Further studies with larger number of premutation carriers will be needed to conclusively identify differences if any, in FMRP levels between males and females.
Point 4: The included participants are mostly male. Were females evenly distributed among probands and non-probands?
Response 4: Yes, females were evenly distributed among probands and non-probands – with 14% of probands being females and 15% of non-probands being females. This difference was not statistically significant (p value = 1.0). This data is presented in Table 1.
Point 5: The statistical comparison of males and females are somewhat difficult to interpret due to the low number of female participants. Do the authors feel that maybe FXAND should be more pronounced in boys compared with girls or is there truely no effect of sex?
Response 5: In general previous studies and our study shows a higher number of males have been probands than females, so this suggests more problems in young male premutation carriers because males have only one X so premutation toxicity would be higher in males than females. We agree, however, that our current study sample did not have adequate number of female participants to draw defintive conclusions regarding prevalence of FXAND in males versus females. We also note that even within the general population, neuropsychiatric disorders such as autism and ADHD are more common in males than females due to various factors. Further studies with larger number of female premutation carriers will be needed to conclusively identify differences if any, in severity and occurrence of FXAND conditions between males and females.
Point 6: Is there any association to number of CGG repeats/FMRP levels - for instance regarding medications or number of FXAND diagnoses or severity of individual diagnoses (proband versus non-proband)?
Response 6: There were no significant associations between number of CGG repeats and number of FXAND diagnoses or number of medications. This could be related to inadequate power to detect differences. Similarly, there were no associations between FMRP levels and the same variables and this is very likely due to the small subset of the sample with FMRP data available.
Point 7: Do the authors have any data on any effects of mosaicism?
Repsonse 7: Mosaicism is largely a feature of individuals with the full Fragile X syndrome where they have methylation mosaicism or CGG repeat size mosaicism. As such, none of the premutation individuals in this cohort had mosaicism; as otherwise they would have been considered to have the full Fragile X syndrome.

Reviewer 2 Report
This article describes the neuropsychiatric profile of people carrying the FMR1 gene premutation in children and young adults. This profile has been studied in adult population, but descriptions in younger ages are few. For this reason, the reviewer has a positive assessment of the topic of the article and believes that it contributes with new and interesting information to previous scientific knowledge.
Taking this into account, there are some aspects that could improve the quality:
- FMR1 gene abbreviation should always be in italics (check text).
- Check punctuation marks.
- In the results section, the following are reported: “Comparing probands and non-pro- bands, there were no statistically significant differences in terms of gender, age and molecular data (CGG repeats, FMRP levels and mRNA levels). However, non-probands had significantly higher FSIQ (99.3 ± 16.6) scores as compared to probands (79.0 ± 25.0; p = 0.001) and higher adaptive behaviour scores as well (VABS ABC 89.8 ±9.1 vs 71.9±14.7, p = 0.004). Scores on the performance IQ (PIQ), verbal IQ (VIQ), VABS Socialization and Daily Living domains were also higher in non-probands as compared to probands (Table 1)”.
These results are very interesting, but are little developed in the discussion section (especially those referring to adaptive behaviour). The reviewer recommends to expand on this point in the discussion.
- The discussion section begins with the sentence “In this study, we demonstrate high rates of FXAND-related conditions….”. Given the small sample size, it is recommended not to use such categorical language and to change it to an expression such as "in this study we have found/observed...".
Author Response
Point 1: FMR1 gene abbreviation should always be in italics (check text).
Response 1: We have checked and ensured this in the revised manuscript
Point 2: Check punctuation marks.
Response 2: These have been checked and corrected as needed throughout the revised manuscript.
Point 3: In the results section, the following are reported: “Comparing probands and non-pro- bands, there were no statistically significant differences in terms of gender, age and molecular data (CGG repeats, FMRP levels and mRNA levels). However, non-probands had significantly higher FSIQ (99.3 ± 16.6) scores as compared to probands (79.0 ± 25.0; p = 0.001) and higher adaptive behaviour scores as well (VABS ABC 89.8 ±9.1 vs 71.9±14.7, p = 0.004). Scores on the performance IQ (PIQ), verbal IQ (VIQ), VABS Socialization and Daily Living domains were also higher in non-probands as compared to probands (Table 1)”.
These results are very interesting, but are little developed in the discussion section (especially those referring to adaptive behaviour). The reviewer recommends to expand on this point in the discussion.
Response 3: This has now been expanded on in the discussion. The revised text reads: ‘Our findings also show that PM carriers as a whole have normal cognition and adaptive behavior scores. This was the case for non-probands who did not have any clinical concerns and were identified through cascade testing. Given that probands were PM carriers who had clinical concerns and hence sought treatment at the center, it is likely that they had some form of impairment in their functioning; this could be related to the presence of ASD which occurred more commonly in probands as well. This would explain the lower IQ and adaptive behavior scores in probands compared to non-probands.’
Point 4: The discussion section begins with the sentence “In this study, we demonstrate high rates of FXAND-related conditions….”. Given the small sample size, it is recommended not to use such categorical language and to change it to an expression such as "in this study we have found/observed...".
Response 4: We have now changed this sentence to not say ‘high’ rates; instead the sentence now reads: In this study, we demonstrate that FXAND-related conditions among young individuals (probands and non-probands) with the PM state occurred frequently, with anxiety and ADHD being the most common.
Reviewer 3 Report
Thank you for your study. It's very necessary to advance in FXS. Numerous psychiatric problems arise with the age of the subjects FSX and others subjects PM. Your paper is very interesting. Except, I think you have a mistake on the line 153. You write: 16 participants, 7 males and 8 females. Perhaps are 15 participants.
Best regards.
Author Response
Point 1: Thank you for your study. It's very necessary to advance in FXS. Numerous psychiatric problems arise with the age of the subjects FSX and others subjects PM. Your paper is very interesting. Except, I think you have a mistake on the line 153. You write: 16 participants, 7 males and 8 females. Perhaps are 15 participants.
Response 1: Thank you for pointing this out and we apologize for the oversight. We did mean 15 participants in all and this has now been changed in the revised manuscript.